# Neuromodulation Techniques in Children with Super-Refractory Status Epilepticus

**DOI:** 10.3390/brainsci13111527

**Published:** 2023-10-30

**Authors:** Ioannis Stavropoulos, Ho Lim Pak, Gonzalo Alarcon, Antonio Valentin

**Affiliations:** 1Department of Clinical Neurophysiology, King’s College Hospital, London SE5 9RS, UK; ioannis.stavropoulos@kcl.ac.uk; 2Department of Basic and Clinical Neuroscience, Institute of Psychiatry, Psychology and Neuroscience, King’s College London, London SE5 8AB, UK; 3Faculty of Life Sciences and Medicine, King’s College London, London SE1 1UL, UK; ho.pak@kcl.ac.uk; 4Royal Manchester Children’s Hospital, Manchester M13 9WL, UK; g.alarcon@nhs.net; 5Alder Hey Children’s Hospital, Liverpool L12 2AP, UK

**Keywords:** super-refractory status epilepticus, neuromodulation, electroconvulsive therapy, deep brain stimulation, vagus nerve stimulation, children epilepsy

## Abstract

Status epilepticus (SE) is a life-threatening condition and medical emergency which can have lifelong consequences, including neuronal death and alteration of neuronal networks, resulting in long-term neurologic and cognitive deficits in children. When standard pharmacological treatment for SE is not successful in controlling seizures, the condition evolves to refractory SE (rSE) and finally to super-refractory SE (srSE) if it exceeds 24 h despite using anaesthetics. In this systematic review, we present literature data on the potential uses of clinical neuromodulation techniques for the management of srSE in children, including electroconvulsive therapy, vagus nerve stimulation, and deep brain stimulation. The evaluation of these techniques is limited by the small number of published paediatric cases (n = 25, one with two techniques) in peer-reviewed articles (n = 18). Although neuromodulation strategies have not been tested through randomised, prospective controlled clinical trials, this review presents the existing data and the potential benefits of neuromodulation therapy, suggesting that these techniques, when available, could be considered at earlier stages within the course of srSE intending to prevent long-term neurologic complications. Clinical trials aiming to establish whether early intervention can prevent long-term sequelae are necessary in order to establish the potential clinical value of neuromodulation techniques for the treatment of srSE in children.

## 1. Introduction

Status epilepticus (SE) has been defined as “a condition characterized by an epileptic seizure that is sufficiently prolonged or repeated at sufficiently brief intervals to produce an unvarying and enduring epileptic condition” [1]. SE is a life-threatening condition and medical emergency, which has an incidence of 14.3–28.4 per 100,000 people per year, affecting all ages, particularly children and the elderly [2]. It is also a condition that, depending on the type and duration of seizures, can have lifelong consequences, including neuronal death and alteration of neuronal networks resulting in long-term neurological and cognitive deficits [1]. A duration longer than 5 min, beyond which, a long seizure or cluster of seizures (without returning to baseline) occurs, is considered as SE. It has been reported in animal models that durations longer than 30 min induce neuronal damage [3], and these timeframes may be variable depending on the type of SE.

Two different types of status epilepticus (SE) have been described: convulsive and nonconvulsive. Convulsive SE shows tonic–clonic, tonic, clonic, or myoclonic manifestations. [4,5,6,7]. Nonconvulsive SE (NCSE) is further classified as generalised, focal, and unknown, depending on the EEG findings. In cases of NCSE with coma and persisting behavioural or awareness changes, the EEG is the most accurate method for appropriate diagnosis [1,8,9].

SE is an emergency condition and guidelines for its management are well elaborated for the choice of first, second-, and third-line drugs of treatment. Benzodiazepines are suggested as the first line of treatment for SE [10]. NICE guidelines suggest intravenous lorazepam, but rectal diazepam or buccal midazolam can also be considered [10,11]. If the SE is not resolved, phenytoin, fosphenytoin, levetiracetam, lacosamide, or phenobarbital can be considered for sustained control. General anaesthesia with propofol, midazolam, or thiopental sodium can be tried in refractory cases of SE (rSE), and super-refractory SE (srSE) is established when it outlasts 24 h of anaesthesia [12,13]. srSE has high mortality and morbidity rates [12,14]. To date, there are no class I data to support recommendations for most antiepileptic drugs for established, refractory, and super-refractory SE.

Timings of SE treatment have been suggested by Shorvon and Ferlisi [12] and Trinka et al. [13], but without clear time recommendations for other types of treatments for rSE. Alternative non-pharmacological techniques have been suggested for rSE in children, including plasmapheresis, ketogenic diet, hypothermia, immunomodulation, and neuromodulation [15,16]. Plasmapheresis is a technique that has been found useful in generalised rSE, largely in adult populations [17]. In paediatric cohorts, only 7 out of 37 children appeared to achieve seizure control [18]. A ketogenic diet has been reported in small case series and larger reports (n = 8–17), showing electrographic seizure resolution within 7 days in 20–90% of patients [15]. A recent clinical trial with hypothermia in an adult population did not find efficacy in srSE compared with placebo [19].

Invasive and non-invasive neuromodulation techniques have been suggested as potential treatments capable of complementing standard pharmacological treatment [20] for srSE. A small number of case reports using neuromodulation techniques have shown some promising results controlling srSE when conventional treatment has failed in children and adults [21].

### 1.1. Non-Invasive Neuromodulation Techniques

Electroconvulsive therapy (ECT) with transcutaneous electrical stimulation of the brain cortex under EEG monitoring is considered a potential treatment for severe major depression and other mental disorders [22] and has also been suggested as a potential treatment for rSE [23]. The technique consists of several sessions of ECT with stimulation intensity and duration parameters, either based on the patient’s seizure threshold or a standard protocol [22,24]. Electrodes can be placed in several positions of the head, either bitemporal, right unilateral (left in left-handed), or bifrontal, depending on the clinical aims [25] (Figure 1A). A serious side effect of ECT is amnesia, retrograde, anterograde, or both, usually improving within 2 weeks [24,26], but a close monitoring of cognitive function is needed to prevent adverse cognitive effects [22]. Details on the technique, the parameters, and the possible side effects have been described in several previous publications [19,20,21,22,23].

Repetitive TMS (rTMS) has recently been considered a diagnostic and potential treatment tool for neurological and psychiatric disorders, including depression, epilepsy, and pain [27,28,29]. rTMS relies on the application of trains of magnetic pulses over the patient’s head, depolarising neurons in the target area [30], and can initially reduce seizures in patients with drug-resistant epilepsy. Transcranial direct current stimulation (tDCS) is a painless, non-invasive stimulation technique that uses polarity-specific electric current to modulate brain excitability and has been used in several conditions [31], including patients with mesial temporal lobe epilepsy [32]. rTMS and tDCS have been reported only for the treatment of srSE in adults and thus, details are not included in the present review.

### 1.2. Invasive Neuromodulation Techniques

Vagal nerve stimulation (VNS) is a NICE-approved procedure for children and adults suffering from drug-resistant epilepsy as an add-on to antiepileptic medication [11]. VNS equipment consists of a VNS pulse generator surgically implanted on the left subclavicular area, including a battery and a 43 cm lead wire with two platinum/iridium helical electrodes (Figure 1B). An external programming system is used to modify stimulation parameters [33,34]. A recent meta-analysis indicates that VNS interrupts srSE in 74% of patients, though the article raises concerns about reporting bias [35]. Reported VNS side effects include dyspnoea, dysphagia, and hoarseness due to vagus nerve damage; bradycardia/asystole during the implantation procedure; postsurgical infections; obstructive sleep apnoea; and tonsillar pain [34,36].

Deep brain stimulation (DBS) includes the implantation of multi-electrode bundles in the brain which are connected to a pulse generator to deliver electrical pulses to modulate the implanted region and functionally connected areas (Figure 1C). Deep brain stimulation is now a technique used worldwide, particularly for the treatment of movement disorders’, but also used in obsessive–compulsive disorder, depression, Tourette syndrome, headache, chronic pain, eating disorders, and epilepsy [37,38,39]. In patients with refractory epilepsy, DBS has been tried for different brain regions, particularly the anterior and the centromedian nucleus of the thalamus [40,41]. The SANTE trial studied the effects of anterior nucleus DBS in patients with focal seizures [41] while centromedian DBS has been mainly tried in patients with generalised epilepsy [42]. Potential DBS side effects include infection, skin erosion, lead migration or fracture, and malfunction of the DBS pulse generator [43]. DBS stimulation-related side effects depend on the area stimulated, the most common being paraesthesia related to DBS intensity [41,44,45].

During the last 4 years, another invasive neurostimulation/neuromodulation has also been described in srSE. This is the responsive neurostimulation (RNS) and consists of depth or subdural electrodes placed in or over one or two predetermined seizure foci. These are connected to a programmable device which is cranially implanted and can provide electrical stimulation in response to detected ictal electrocorticographic activity [46]. About 10 cases have been described and the results are positive [47,48,49,50], and likely more cases will be described soon, but the published peer-reviewed cases regard adults and thus further details were considered out of scope of this review for the paediatric population.

Although neuromodulation has potential in the management of epilepsy and srSE, there have been only a few studies in children showing the potential benefits of these techniques after conventional medical treatment for srSE has failed. This systematic review aims at evaluating the potential benefits of neuromodulation for srSE in the paediatric population.

## 2. Materials and Methods

This systematic review was performed in line with the Preferred Reporting Items for Systematic Reviews and Meta-Analyses (PRISMA) Guidelines [51]. A comprehensive literature review was performed using PubMed and MEDLINE to find relevant articles published from 1946 to August 2022. The selection criteria included all relevant subject headings and freeform texts relating to neuromodulation techniques and paediatric status epilepticus. The following search strategy for PubMed was performed on 15 September 2021: (VNS OR vagal nerve stimulation OR vagus nerve stimulation) OR (transcranial magnetic stimulation OR TMS) OR (electroconvulsive therapy OR ECT) OR (deep brain stimulation OR DBS)) AND ((status epilepticus OR epilepsia partialis continua OR refractory status epilepticus OR rSE OR super-refractory status epilepticus OR srSE) OR (“Febrile infection-related epilepsy syndrome” OR “FIRES”) OR (“New-onset refractory status epilepticus” OR “NORSE”)). A similar search strategy was used for MEDLINE using subject headings and freeform text on the same day. All articles were imported for screening to Covidence (© Cochrane). Duplicates were automatically removed before initial screening by the software. Title and abstract screening and full-text assessment were performed independently by two reviewers (H.L.P. and A.V.). Articles in English referring to neuromodulation (VNS, TMS, ECT, or DBS) and status epilepticus lasting more than 24 h with anaesthesia in patients <18 years were included. Original articles, case studies, case reports, and letters to the editor were included while conference articles, literature reviews, and systematic reviews were excluded. Conflicts in screening and full-text assessment were resolved by three reviewers (H.L.P., A.V., and I.S.).

Studies should give an estimation of the number of days between SE onset and initiation of neuromodulation therapy, an estimation of the number of days from neuromodulation to any changes in patient condition, and the final results of the neuromodulation therapy. Neuromodulation therapy was regarded as successful when it led to cessation of both clinical and electrographic SE, and when the patient was stable enough to be transferred out of ICU. This data was then converted into graphs to illustrate the timeline for all relevant patients. Apart from the timeline details, data included demographics (patient’s age and gender), type of neuromodulation, and epilepsy before the onset of SE.

## 3. Results

The Prisma flowchart is presented in Figure 2. After excluding duplications and papers not fulfilling the inclusion criteria, 18 references were included for further analysis in the present review. The neuromodulation techniques reviewed were VNS (n = 15), DBS (n = 6), and ECT (n = 5). The mean age of the patients was 7.4 years, ranging from 0.5 to 17 years. Twelve out of twenty-five patients were female.

Basic demographics of patients are illustrated in Table 1 and Table 2. Different neuromodulation protocols for srSE in children have been described, with heterogeneous approaches for results description.

### 3.1. Electroconvulsive Technique (ECT)

ECT was used in five patients reported in five publications (Figure 3) [53,59,60,61,62]. srSE was focal in all patients, showing secondary generalisation in two. In three patients, srSE occurred de novo and in two was due to FIRES. MRI was not reported in one patient, showed bilateral polymicrogyria in one, and was initially normal in three (and later in the course of the srSE showed atrophy in one patient). ECT was applied between bitemporal electrodes in two patients and between frontotemporal electrodes in two patients, and electrode positions were not specified in one patient. The number and frequency of ECT sessions varied as seizure control was attempted. The number of days that each patient had been on srSE before ECT was started was 14, 24, 50, 60, and 120 days. Three patients had ECT initially on a number of consecutive days (7 or 12), one had five ECT sessions within 9 days whereas the fifth patient had ECT in two pairs of two consecutive days each, separated by 5 days. The number of days of ECT before srSE remission was 2, 5 (in two cases), 7, and 12 days. srSE stopped on the last day of ECT in three patients, while in two patients srSE stopped 4 days before the end of ECT was completed (after 29 and 125 days of SE). Therefore, ECT was considered to have contributed to stopping srSE in all five reported cases. Two patients remained out of the srSE without sequelae (one had learning difficulties beforehand), one had right temporal lobe surgery for focal cortical dysplasia, one developed severe epileptic encephalopathy, and one remained seizure-free but with severe motor dysfunction and cognitive decline.

### 3.2. Vagus Nerve Stimulation (VNS)

Fifteen patients had VNS implantations for srSE in nine articles (Figure 4) [52,54,55,56,57,58,63,64,65]. In three patients, srSE occurred de novo due to FIRES. srSE was focal or secondarily generalised in 10 patients, myoclonic in one, spasms in two, primarily generalised (tonic, T-C, myoclonic, absence seizures) in one, and GTCS without further explanation in one [65]. Identified aetiologies among the focal srSE included malignant partial epilepsy of infancy in four patients (in one patient due to mother’s heroin abuse in pregnancy), neonatal venous thrombosis in one patient, FIRES in three, hemimegalencephaly in one, and one patient had bilateral frontal simplification of cortical gyri together with progressive diffuse cerebral atrophy. Spasms were due to non-ketotic hyperglycinemia in one patient and to microdeletion of 1q43q44 in another patient. Head MRI was normal in seven patients and showed cerebral atrophy/microcephaly in four patients, hemimegalencephaly in one, thalamic lesion/stroke in one, and there was no information about imaging in two patients. There was significant heterogeneity among the VNS parameters. The amplitude varied from 0.25 mA to 3 mA, usually with progressive increments during the course of the treatment until srSE improvement was achieved. The “on” time was 7 s in one case, 14 s in another case, and 30 s in the remaining patients. The “off” period was between 1.8 and 5 min.

Among the fifteen patients with VNS, the srSE was resolved in twelve. Two patients recovered without seizures, seven recovered from srSE remaining with seizures and learning difficulties due to underlying conditions, and three patients recovered but died of unrelated causes (dilated cardiomyopathy 5 months late, paediatric acute respiratory distress syndrome 2.5 years later, and tracheostomy-related late bleeding). Three patients continued with srSE after VNS implantation (two died and one was implanted with CMN DBS).

### 3.3. Deep Brain Stimulation (DBS)

DBS was performed on six patients reported in five articles (Figure 5) [52,66,67,68,69]. In four patients, srSE occurred de novo, due to FIRES in three cases. The centromedian nucleus was stimulated in five patients and the anterior nucleus in one. srSE was focal in five patients, requiring intubation and induced coma in one, and generalised tonic–clonic in one. Head MRI was reported in five patients and was normal in three, showed severe cytotoxic oedema in one, and another patient showed signal abnormalities in basal ganglia, external capsule, and cortex. Four patients had new onset SE without a previous history of seizures or epilepsy (de novo), including three patients where srSE followed febrile illness, suggesting FIRES. Four patients underwent trials of cessation of the stimulation after an initial improvement (stimulation period A) in order to demonstrate the stimulation effect. As SE returned, stimulation was then re-started (stimulation period B). The duration of stimulation period A was 15, 18, 22, and 67 days. After stimulation period B, patients were discharged with the stimulator on. The stimulation frequencies were 6 Hz (in three patients), 145 Hz, and 180 Hz. DBS had immediate effects on srSE in two patients. In the remaining four children, improvement occurred after 2, 4, 6, and 30 days of stimulation. SrSE resolved in all patients; one patient came back to her previous number of seizures, four children remained with seizures, and one child remained in a vegetative state.

In summary, 23 out of the 25 children treated with neuromodulation techniques recovered from srSE (Table 2). Among these children, twelve recovered without new medical conditions, four recovered with severe sequelae (epileptic encephalopathy or cognitive/motor decline), four developed new seizures after the srSE, and three died for unrelated reasons.

## 4. Discussion

Status epilepticus (SE) is a medical emergency with high mortality and morbidity rates [70,71,72]. The treatment protocols for the early management of SE are well standardised [12]. However, refractory and super-refractory SE (rSE and srSE) are often associated with significant and irreversible brain damage whose severity is related to SE duration and aetiology. A well-defined, effective, and fast-acting therapeutic protocol would be highly desirable to prevent potential longstanding neurologic complications. In this review, we present data showing that neuromodulation could be a potentially efficacious treatment option for shortening rSE and srSE duration in children when the routinely used 1st, 2nd, and 3rd line treatments have failed.

The features of status epilepticus in children can be slightly different from those in adults. As shown in the presented data, srSE in children is often caused by genetic/metabolic conditions, brain malformations, birth injuries, and febrile infection-related epilepsy syndrome (FIRES). The latter is a rare, life-threatening condition that presents with a non-specific febrile illness followed by refractory status epilepticus within 24 h to 2 weeks of the onset of the febrile illness in previously healthy children, with a mortality of up to 30% [73]. Its pathogenesis is unclear, but autoimmune mechanisms have been proposed [74], and a recent international consensus recommendation suggests that first-line immunological treatment should be started during the first 72 h [75].

There are no clinical trials performed to assess the efficacy of any of the neuromodulation techniques on srSE. The literature shows that three different neuromodulation techniques, one non-invasive (ECT) and two implantable devices (VNS and DBS), have been sporadically tried and could be beneficial in cases of rSE and srSE in children. However, the number of reported cases remains small, there is significant diversity regarding the cause of the srSE, and the mechanisms by which neuromodulation affects SE are not elucidated. DBS and ECT appear to have provided benefit in cases with the FIRES condition, while only one out of the three cases published with FIRES and VNS showed resolution of the srSE. All three techniques share unclear mechanisms of action. Animal studies suggest that ECT alters biological processes such as neuroplasticity and neurotransmitter function and might cause internalisation of NMDA receptors or other epigenetic effects [76,77].

Several studies suggest that VNS can modify norepinephrine and serotonin levels at the locus coeruleus and dorsal raphe nuclei [78]. It has also been suggested that VNS can cause changes in limbic structures’ functions modifying GABA and glutamate concentrations at nucleus tractus solitaries [79]. Similar mechanisms may be effective against srSE.

The main advantage of DBS is that the electrical stimulation can be applied locally to specific brain areas using implanted intracranial electrodes and that different stimulation parameters can be applied at the implanted region. In some published cases, electrode implantation probably induced a microlesion effect which was associated with major seizure improvement [39,80]. It has also been proposed that upon high-frequency stimulation (>60 Hz), inhibition of the stimulated area might be mediated by activation of GABAergic afferents or inactivation of voltage-gated currents [39,81]. Moreover, low-frequency stimulation (6 Hz) of the centromedian thalamic nucleus has recently been reported to be useful in reducing the severity and frequency of focal seizures in children and adults with srSE [66,67,82], probably via neuromodulation of cortical structures through the thalamocortical pathway [83].

At present, there is no consensus protocol for the use of ECT, VNS, and DBS in the management of rSE/srSE, and existing evidence is based on a limited number of reported patients. Not all neuromodulation techniques are clinically available in most centres, and they are only considered at late stages of rSE/srSE when standard treatment has failed. As brain damage caused by SE can start as early as 30 min from SE onset [3], the appropriate time for the application of neuromodulation for the treatment of srSE in children is a question of major importance.

Even though published paediatric cases show that neuromodulation was applied as a last-resort treatment, the results appear encouraging. Neuromodulation techniques were applied between 5 [56] and 120 days [59] after SE onset, and substantial brain damage may have already been present in most cases by that time. Regarding the use of non-invasive neuromodulation techniques, the use of ECT in children has given limited but promising results, suggesting that the non-invasive safer techniques could be considered earlier in the course of srSE. Despite the side effects and the invasive nature of VNS and DBS, such damage is unlikely to be induced by neuromodulation, as similar techniques are safe when used for the chronic treatment of epilepsy or other brain conditions [22,39,84,85]. Unfortunately, differences in the timing to apply neuromodulation techniques in different centres/cases do not allow reliable conclusions on the optimal timing for starting this treatment modality.

Regarding invasive procedures, the present series suggests that the effects of DBS and VNS can occur within the first week of treatment. Nonetheless, DBS was used in fewer children, and VNS appears to have been used in a higher number of patients with severe epilepsy (migrating epilepsy, severe cortical malformations, birth injury). DBS implantation led to improvement of srSE, with worsening in seizures when the DBS was turned off [66,67,69,82,86].

Even though the presented data look encouraging, clinical guidance cannot be based on published case reports due to the risk of significant bias [87]. For instance, it is common to find successful neuromodulation cases for srSE published as single case reports, but unsuccessful cases are usually published as part of a case series [56,88] or are not submitted for publication.

As suggested by Rossetti and Lowenstein [20], neuromodulation could be complementary to pharmacological treatment for the management of rSE. Non-invasive techniques such as ECT, tDCS, or rTMS could be considered as add-on treatments after the failure of standard treatment. If no improvement is noted, invasive techniques (DBS and VNS), when available, could be discussed and planned in a timely manner.

## 5. Conclusions

The evaluation of neuromodulation techniques for the treatment of srSE in children is limited by the small number of published cases and the variability of neuromodulation protocols used for the treatment of srSE. Although neuromodulation strategies have not been tested through randomised, prospective controlled clinical trials, this review presents the existing data and the potential benefits of neuromodulation therapy, suggesting that these techniques could be considered at earlier stages within the course of srSE intending to prevent long-term neurologic complications. Clinical trials aiming to establish whether early intervention can prevent long-term sequelae are necessary to establish the potential clinical value of neuromodulation techniques for the treatment of srSE in children.

## Figures and Tables

**Figure 1 brainsci-13-01527-f001:**
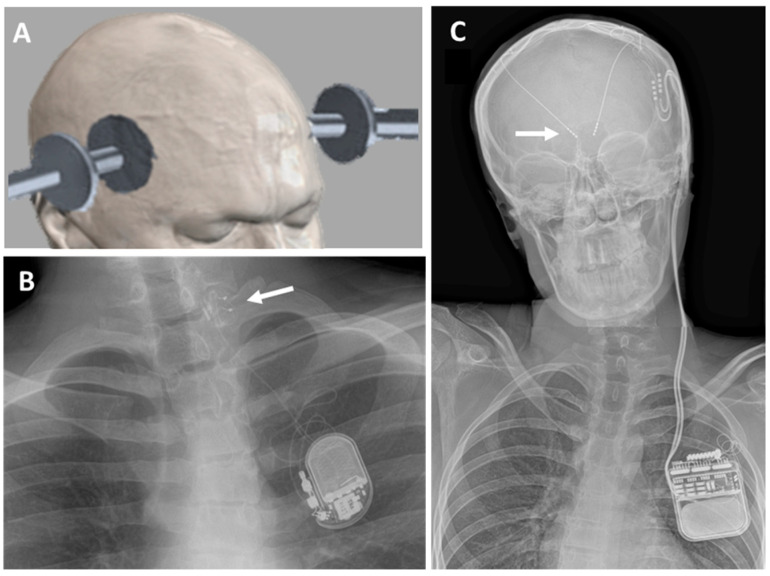
Invasive and non-invasive neuromodulation techniques tried in children with srSE. (**A**) Electrodes and typical positions for non-invasive electroconvulsive therapy (ECT); (**B**) X-ray showing vagus nerve stimulation (VNS) in a 14 y/o child. The white arrow indicates the position of the stimulating contacts in the vagus nerve; (**C**) X-ray showing deep brain stimulation (DBS) in a 12 y/o child. The white arrow indicates the DBS position in the centromedian thalamic nucleus in the brain.

**Figure 2 brainsci-13-01527-f002:**
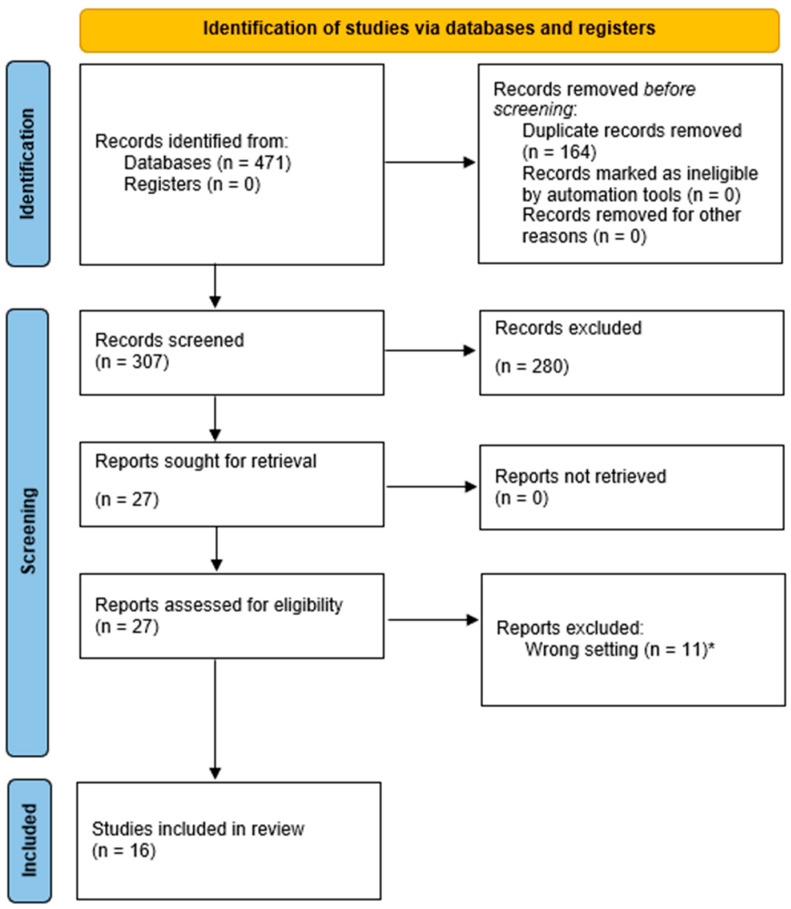
PRISMA 2020 flow diagram for new systematic reviews. * 6 studies reported epilepsia partialis continua, 1 study reported asleep electrographic status epilepticus, 1 study reported Rasmussen’s encephalitis, and 3 studies where neuromodulation did not directly treat the status epilepticus.

**Figure 3 brainsci-13-01527-f003:**
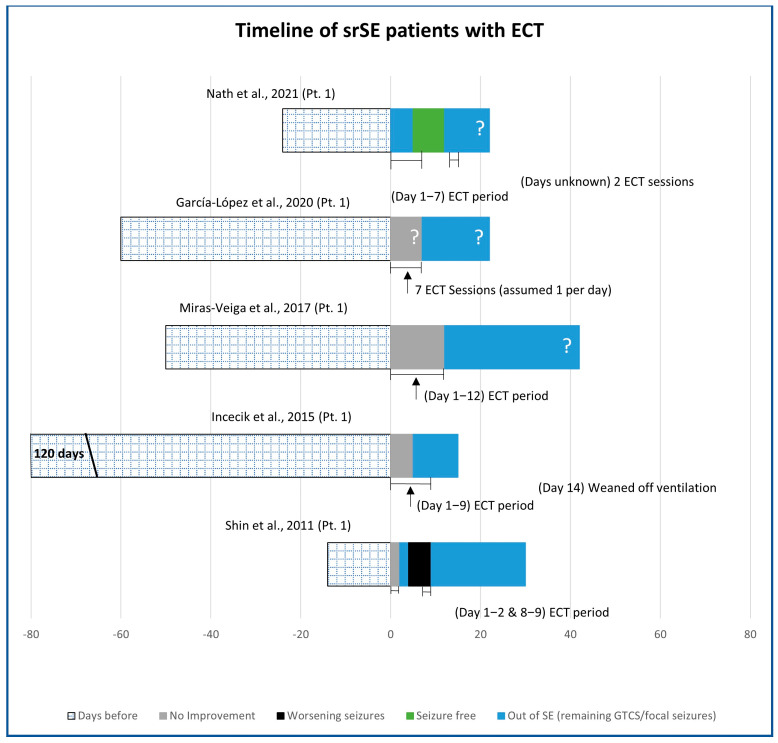
Timeline of srSE patients treated with ECT. (? = the time period is estimated but not clearly stated) [53,59,60,61,62].

**Figure 4 brainsci-13-01527-f004:**
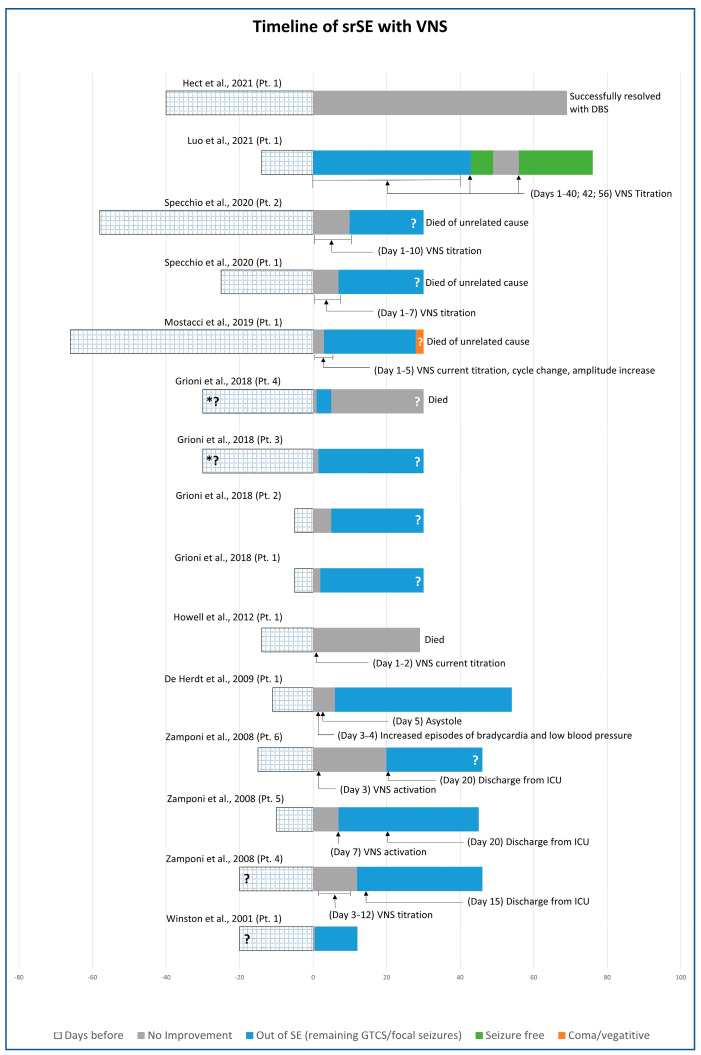
Timeline of srSE patients treated with VNS (* about 30 days but not clearly stated; ? = the time period is estimated but not clearly stated) [52,54,55,56,57,58,63,64,65].

**Figure 5 brainsci-13-01527-f005:**
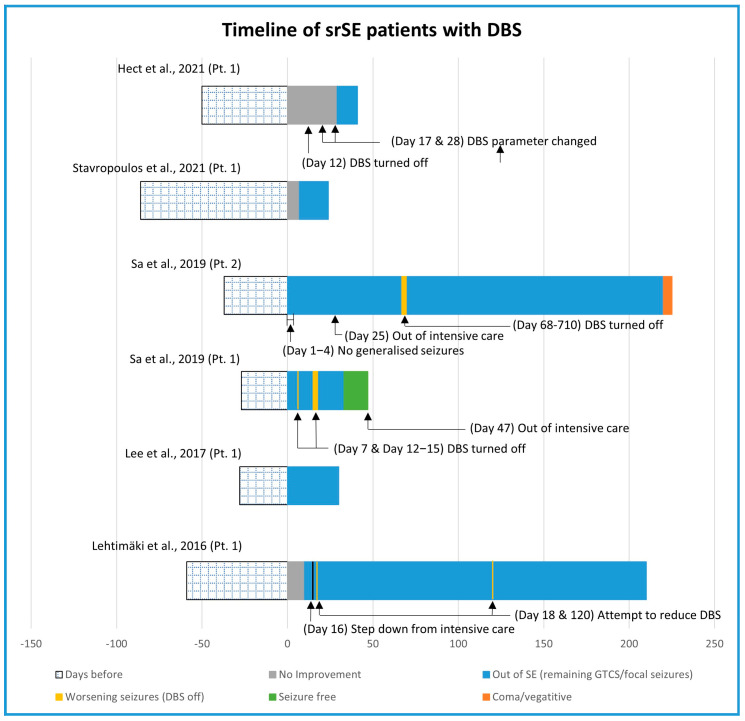
Timeline of srSE patients treated with DBS [52,66,67,68,69].

**Table 1 brainsci-13-01527-t001:** Basic demographic including age, gender, and type of neuromodulation. * One patient [52] was reported in both VNS and DBS as the patient underwent both neuromodulation techniques. VNS did not offer benefits, and DBS caused resolution of super-refractory status epilepticus.

	Number of Patients	Male	Female	Stimulation Age (Years)	De Novo	Post Febrile
ECT	5	2	3	6.8 (3^−^–−16)	3	2
VNS	15 *	8	7	5.73 (0.5–16)	3	3
DBS	6 *	3	3	12.5 (5–17)	4	3
All	25	13	12	7.4 (0.5–17)	9	7

**Table 2 brainsci-13-01527-t002:** Outcomes following neuromodulation. Stim = stimulation; SE = status epilepticus. * One patient [53] was removed from calculation of “Duration of stimulation before SE resolution” as it was not adequately reported. ** Four patients ([54]; [pt 4] [55]; [pt 3 and pt 4] [56]) were removed from calculation of “Duration of SE before stimulation” as it was not adequately reported. *** 3 patients died of unrelated causes after SE resolved ([57]; [pt 1 and pt 2] [58]).

	SE Duration before Stim (Days)	Stim Duration before SE Resolution (Days)	Recovered from SE	Severe Sequelae Post-SE	Died during SE
ECT	53.63 (14–120) N = 5	4.75 (0–12) * N = 4	5	3	0
VNS	23.9 (5–66) N = 11 **	6.2 (0–20) N = 12	12	0	2 ***
DBS	49 (27–86) N = 6	7.7 (0–29) N = 6	6	1	0
All	37.5 (5–120) N = 21	6.3 (0–29) * N = 22	23	4	2

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
