# Peer review of "Neuromodulation Techniques in Children with Super-Refractory Status Epilepticus"

_brainsci, 2023, doi:10.3390/brainsci13111527_

Round 1

Reviewer 1 Report

Comments and Suggestions for Authors

This work presents the current state and role of neurostimulation therapies in the context of super refractory status epilepticus in the pediatric population. The authors have focused on both non-invasive (ECT, rTMS) and invasive (VNS, DBS) techniques and evaluate the current published literature. Their analysis suggests that early application of neurostimulation appears to be reducing the risk of long-term neurological complications and suggest more systematic research to be focused on further quantifying the efficacy of neurostimulation. The Paper is well organized, cites and discusses the relevant literature appropriately. The methods are well-described, and the results are presented in a clear manner. I am happy to endorse the paper for publication, after a minor addition for completeness purposes:

- This review paper is addressing the non-US epilepsy community, in the sense that the authors did not include the application the FDA-approved Responsive Neurostimulation (RNS) in srSE. There is growing literature that suggests responsiveness in srSE either with RNS alone, (e.g. Ernst et al., J Neurosurg, 2023; Ryvlin and Jehi, 2021) or combined with resective approaches, (e.g. Mamaril-Davis et al., World Neurosurg 2022), especially when patients are refractory to other neurostimulation approaches (Yang et al., Ann Clin Transl Neurol. 2021). Although most of these were applied in adult populations, I strongly believe that this growing field deserves mention and productive discussion in a separate paragraph in the Discussion section.

Author Response

Thank you very much for your comments. As suggested, we have included a new paragraph in the article regarding RNS.

During the last 4 years, another invasive neurostimulation/ neuromodulation has also been described in srSE. This is the Responsive Neurostimulation (RNS) and consists of depth or subdural electrodes placed in or over one or two predetermined seizure foci. These are connected to a programmable device which is cranially implanted and can provide electrical stimulation in response to detected ictal electrocorticographic activity [45]. About 10 cases have been described and the results are positive [46-49] and likely more cases will be described soon but the published cases in peer reviewed regard adults and thus further details were considered out of scope of this review for the paediatric population.

Reviewer 2 Report

Comments and Suggestions for Authors

This systematic review focused on Status Epilepticus (SE). The purpose of the review was to summarize the rather scant existing data and benefits of neuromodulation strategies. The review focused on electroconvulsive therapy, deep brain stimulation, and vagus nerve stimulation. Repetitive TMS and tDCS were not covered. A total of 25 cases were evaluated in a total of 18 articles. The overall conclusion of the review was that these methods show promise in the available literature, but all of the studies have essentially been case studies and clinical trials are needed, which will be challenging.

Overall, this was a good review and on an important topic that has not received a large amount of research focus. The review overall was well-written and very good. There were very few grammatical errors, typos, or English errors, which is rare for papers I review nowadays. I don’t think the paper had any weaknesses other than the obvious limitations (low number of total subjects, low number of studies, case studies, etc) that the authors acknowledged. Therefore, I only have some minor comments the authors should address before the study can be published.

-Figure 1 I think there are still some track changes marking in it in the screening section.

-Figures 2-4 especially Figure 3: If possible can some of the fonts be increased in size, they are somewhat small.

-Line 300. No space after the period after “[72]”

-Bibliography has some errors. Some titles of articles the first letter of each word is capitalized, whereas others are not. Please proofread and check the bibliography extensively.

Author Response

Thank you very much for your comments. We have corrected the mistakes as follows:

-Figure 1 I think there are still some track changes marking in it in the screening section. Corrected

-Figures 2-4 especially Figure 3: If possible can some of the fonts be increased in size, they are somewhat small. The fonts have been increased, if still considered small they could be converted to a larger font

-Line 300. No space after the period after “[72]” Thanks, corrected.

-Bibliography has some errors. Some titles of articles the first letter of each word is capitalized, whereas others are not. Please proofread and check the bibliography extensively. We have checked the bibliography and we have corrected this mistake

Reviewer 3 Report

Comments and Suggestions for Authors

Refractory status epilepticus is the most important and serious problem in both adult and pediatric epileptology. This is associated with a high risk of death and a high percentage of neurological deficits that persist after relief of status epilepticus. In this regard, the search for additional methods to relieve this acute and life-threatening condition is very relevant. Thus, the relevance of the manuscript in question is beyond doubt. During the audit, a number of comments arose:

1) It makes sense for the authors to expand sections 1.1, 1.2 and 1.3, supplementing them with a detailed description of the methods under consideration (drawings or diagrams). This is necessary for a clearer understanding of the manuscript by neurologists not specializing in epileptology.

2) In the “Results” section, authors must provide a comparative description of the results of using all three methods in the form of a table or figure.

3) The list of references includes 77 sources, but most of the sources are very outdated. Authors should add references to recent literature sources.

Author Response

Thank you very much for your interesting comments, that can make the article more helpful for general neurologists. We have made the following changes:

1) It makes sense for the authors to expand sections 1.1, 1.2 and 1.3, supplementing them with a detailed description of the methods under consideration (drawings or diagrams). This is necessary for a clearer understanding of the manuscript by neurologists not specializing in epileptology. Following your advice, we have included a figure showing the different techniques reviewed in this article (DBS, VNS and ECT). 

2) In the “Results” section, authors must provide a comparative description of the results of using all three methods in the form of a table or figure.

We have added Table 2 as a comparative description of the different methods

3) The list of references includes 77 sources, but most of the sources are very outdated. Authors should add references to recent literature sources.

Thank you for this suggestion. We have included new references regarding RNS and SE in some sentences (for instance: Becket et al 2023; Cruickshank et al 2022 and Cornwall et al 2023).  Some relatively old references are kept in the manuscript as they can have historical significance. If newer references are suggested for any particular section, we will be delighted to include them in the article.